# Radial bimetallic structures via wire arc directed energy deposition-based additive manufacturing

Lile Squires [1], Ethan Roberts [1] & Amit Bandyopadhyay [1] ✉

Bimetallic wire arc additive manufacturing (AM) has traditionally been limited to depositions characterized by single planar interfaces. This study demonstrates a more complex radial interface concept, with in situ mechanical interlocking and as-built properties suggesting a prestressed compressive effect. A 308 L stainless core is surrounded by a mild steel casing, incrementally maintaining the interface throughout the Z-direction. A small difference in the thermal expansion coefficient between these steels creates residual stresses at their interface. X-ray diffraction analysis confirms phase purity and microstructural characterization reveals columnar grain growth independent of layer transitions. Hardness values are consistent with thermal dissipation characteristics, and the compressive strength of the bimetallic structures shows a 33% to 42% improvement over monolithic controls. Our results demonstrate that biomimetic radial bimetallic variation is feasible with improved mechanical response over monolithic compositions, providing a basis for advanced structural design and implementation using arc-based metal AM.

The fascinating complexities in naturally occurring structures serve as the basis for many modern engineering designs, with scientific thought and technological advancement perpetually reaching an ever-greater understanding of natural mechanisms[1]. Recognition and identification of natural design principles inevitably expand performance capability and functionality aspirations[2,3]. One such concept that immediately draws attention is the prevalence of multi-material structural designs found in nature. Rarely is a natural system composed of only a single material, and even primarily homogenous systems are likely to have minor materials involved. When a natural multi-material structure – cancellous bone embedded within cortical bone, for instance – is studied, it is noted that the arrangement of material variation responsible for mechanical properties is rarely along a single plane[4,5]. More common is axial variation, with inner layers surrounded by outer layers, each region of material lending unique performance capability to the function of the whole (Fig. 1). In wood anatomy, this relationship is easily recognized in the concentric rings commonly exposed with cross-grain cuts. From the outer living phloem to the innermost heartwood, each region plays a function vital to the overall survival of a tree, with the interaction between regions responsible in part for the flexible yet mechanically robust structure[6,7]. Performance gains from such multi-material arrangements draw attention in the scientific community, with attempts to imitate those gains through increasingly challenging biomimetic designs. For example, in the construction of metallic replacements for bone, a focus on the chemistry or development of monolithic porous structures has been encouraging but still leaves much to be desired[8,9]. Progress will continue from a design perspective only by integrating those approaches with a compositional variation.

Additive manufacturing (AM) technology is particularly suited to maximum design flexibility in compositional variation. A recent focus by industry and academia on this technology – and metal AM – is already pushing the operational possibilities forward. Medical, transportation, energy, and aerospace industries have all

[1]W. M. Keck Biomedical Materials Research Lab, School of Mechanical and Materials Engineering, Washington State University, Pullman, WA 99164-2920, USA. ✉e-mail: amitband@wsu.edu

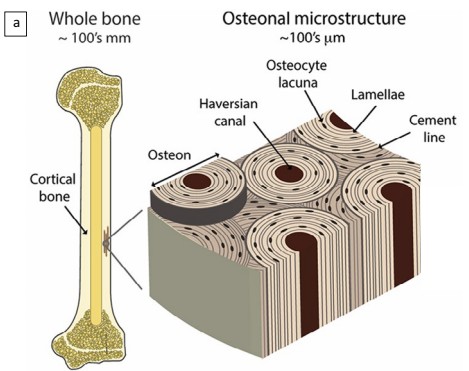
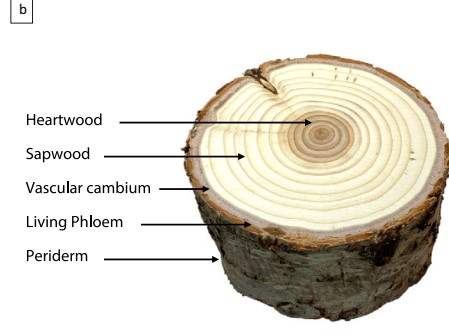

Fig. 1 | **Natural radial structures. a** Variation in compositional architecture is exhibited in natural multi-material structures such as bone. Variation is not observed along a single plane, but is found in complex structural and material arrangements that largely govern mechanical performance and unique functional properties. Adapted from Zimmermann et al.[5]; **b** Annular architectural variations in wood anatomy is easily recognized in the concentric rings commonly exposed with cross-grain cuts. Each region fills a function vital to the overall survival of a tree, with the interaction between regions responsible in part for a flexible yet strong structure.

jumped at the rapidly expanding technology and are seeking to take advantage of the flexible capabilities offered[10]. Capability for complex shapes and designs, including thin-wall cylinder geometry, is found in conventional casting methods, but design changes equate to expensive tooling modifications. AM is fundamentally a tool-free manufacturing approach, where design changes or variations are easily incorporated. Of the various metal AM technologies, research primarily focuses on powder-based directed energy deposition (DED) and powder bed fusion (PBF) techniques using lasers or electron beams, or on wire-feedstock arc-driven methods using MIG, TIG, or plasma arc welding[11]. Sophisticated design and build strategies for metal AM produce innovative lightweight, high-performance components for aerospace and automotive industries and custom-tailored medical implants, among many other exciting applications[12].

In nearly every case, AM produces structures in ways beyond the capability of traditional manufacturing methods[13]. At the same time, most investigations are confined to single material compositions – and metal AM is well positioned to produce complex multi-material systems[14]. The versatility of bimetallic-capable additive technology expands the design space by exploiting unique constituent material characteristics in as-built structures[15,16]. Powder-based DED is particularly suitable for creating such bimetallic structures, with various powder compositional variants easily created to promote specifically designed functionalities[17–19]. For instance, layers of one material may impart corrosion resistance to a structure, while layers of another material may contribute high tensile strength within that same structure. Powder-based metal AM has a proven record with commercially produced equipment, making complex parts with established post-processing steps. These parts are produced in enclosed inert chambers, as powders are melted together layer by layer during precise scanning of the energy beam. Part size, however, is limited by the build chamber dimensions. The pinpoint scanning process of powder DED also hamstrings production with low deposition and yield rates[20]. Another option for bimetallic deposition is wire arc AM (WAAM, also known as DED-arc and WA-DED), a popular additive technology actively transitioning from a research focus to commercialized use. It is primarily motivated by low equipment costs, high deposition rates, and theoretically unlimited build volumes[16,21]. Research advancement in DED-arc technology is focused on an increased understanding of process parameters and their influence on material properties. Part quality improvement is anticipated through hybrid machining capability, improved path planning, parameter optimization, thermal management, and deposition mechanics[22,23]. Here also, bimetallic optimization in material type and

application-specific material combinations are beginning to be addressed, promoted by the consistent and reliable deposition of wire-based feedstock[14,15,17,24–27].

Interestingly, it appears that bimetallic DED-arc research is most commonly explored in depositions characterized by horizontal compositional variation or single interfaces in the vertical orientation[15,17,24–26,28–30]. Representative of the many studies, Ahsan et al. demonstrate successful multi-material DED-arc functionality in depositing low-carbon steel on top of stainless steel in a thin-walled structure[24]. Their analysis shows no weld defects, with the interface between the dissimilar metals characterized by a diffusion of chromium and increased hardness, resulting in decreased ductility. Other researchers have comparable results when investigating the use of bimetallic combinations in stacked deposition structures[16,21]. These bimetallic interfaces are relatively straightforward, aligned vertically along the Z-axis or layered one on top of the other. Greater complexity along the horizontal build path with dissimilar metals using an interweaving deposition pattern is also attempted with some success[31]. The mechanical behavior of the resultant bimetallic structure is notably improved, with solid solution strengthening observed. In addition to stacked bimetallic structures and interweaved deposition bimetallic structures, the in situ weld pool mixing of dissimilar metals is also investigated; Huang et al. produce a functional gradient across the deposition layer, with the arc stirring various ratios of dissimilar filler wire[30]. Even so, attempts to create multi-dimensional bimetallic structures with complex xy plane radial interfaces established and maintained vertically throughout the structure in the Z-axis are not seen.

In this work, we investigate the feasibility of multi-dimensional bimetallic structures with xy plane variation maintained throughout Z-axis layering, to advance the concept of bimetallic DED-arc (Fig. 2). A multi-torch DED-arc system is developed and integrated, capable of creating and maintaining radial bimetallic compositional interfaces with simultaneous or sequential deposition. We also seek to understand how annular compositional variation can change overall performance when a small effective variation in the coefficient of thermal expansion (CTE) is present between the two materials in a multi-material structure. To characterize the effect of residual internal stresses between components arranged in this fashion, 308 L stainless and mild steel are used as the bimetallic couple. Samples are deposited layer by layer (Fig. 2c) in three different configurations: solid 308 L stainless steel, solid mild steel, and radial bimetallic. The latter comprises a central stainless-steel core surrounded by a low-carbon steel outer casing (Fig. 2d–f). Using these two well-understood materials focuses attention on the bimetallic process and promotes parametric

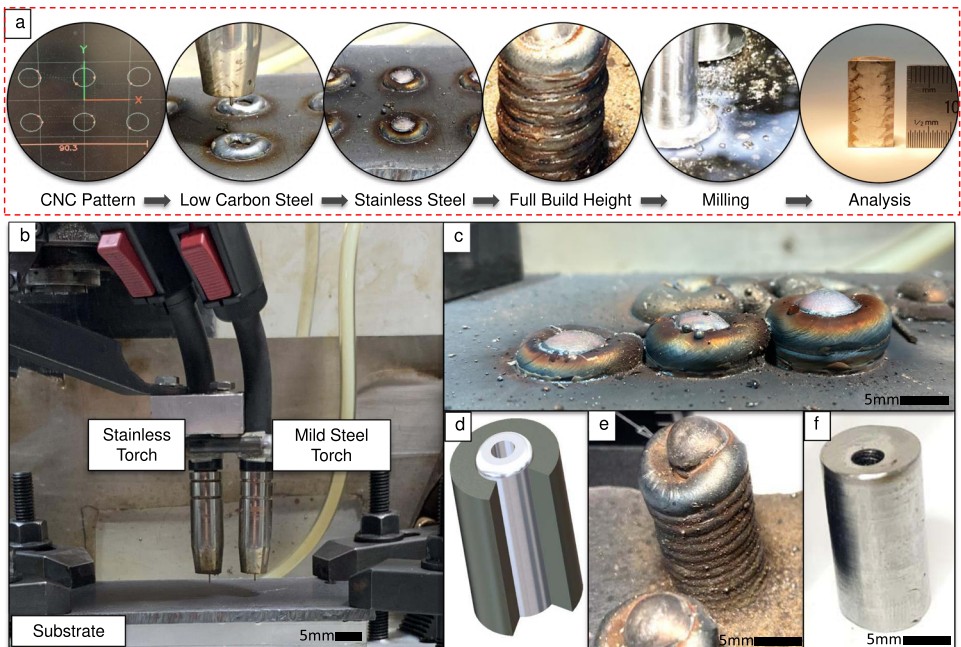

**Fig. 2 | Radial bimetallic DED-arc deposition. a** Process sequence flow chart for bimetallic DED-arc with a radial variation. Deposition path planning is performed using software, then output to the machine tool. The mild steel casing is deposited first, forming a circular bead around a hollow depression. 308L stainless steel is immediately deposited into that hollow, and the process repeats with each layer stacking on top of the previous until full build height is reached. Milling is performed on the depositions until desired final dimensions are obtained. Separation from the base plate allows analysis and testing; **b** Arrangement of a representative dual-mount torch setup in a CNC mill. Fixed to the CNC spindle, both torches are independently activated through CNC code and follow separately programmed deposition paths with feedstock supplied by independent welding power sources; **c** Deposition sequence illustrating parallel build deposition for inner and outer materials, with progressive maintenance of the bimetallic interface upward in the Z build direction; **d** Conceptual 3D model for a bimetallic corrosion-resistant tube structure, consisting of a stainless core with mild steel casing; **e** Completed deposition pillar used to produce radial bimetallic concept structures; **f** Machined and drilled radially bimetallic concept structure.

learning that benefits the investigation of more advanced materials. Quasi-static compression testing is performed, the microstructure and phases are analyzed, and hardness values are measured. Process complexity from the dual deposition of different metals on a single additive layer is discussed, including added depositional thermal input, exacerbated layer height concerns, interface mechanics, and the role of deposition path planning[30,32,33]. The ramifications of these process inputs on interface formation, microstructure, and mechanical performance are of critical interest in this work and form a foundational theory for multi-dimensional radial bimetallic arc-based additive manufacturing.

## Results

The cumulative effects of radial combination along the Z-axis build direction for the two materials via wire arc additive processing were evaluated through SEM-EDS and microstructural imaging (Fig. 3). XRD and hardness analysis was completed, as well as investigation of compressive deformation and material flow constraint through the use of a drilled passage (Fig. 4). Compression testing was performed on all specimen types, with optical analysis of deformation severity (Fig. 5) and interface fusion (Fig. 6). Interface textures in a non-deformed bimetallic specimen were further characterized through EBSD (Fig. 6d–g). Residual pressure experienced in the bimetallic couple during cooling was explored through modeling and additional compressive tests (Fig. 7).

### Microstructure and phase analysis

Microstructural imaging reveals fine equiaxed grain formation in the mild steel casing and the stainless core (Fig. 3). A distinct interlocking pattern between the bimetallic couple is evident in the cross-sectioned specimens and can be seen clearly in Fig. 3b. EDS of the interface between the stainless core and the mild steel casing reveals slight migration of Cr and Ni from the core into the casing (Fig. 3c.1–c.3). At transition zones between successive layers in the stainless core, columnar grain growth independent of layer boundaries is observed, oriented with the build direction in the Z-axis (Fig. 3g). Phase analysis reveals the presence of all expected peaks (Fig. 4d), measured across the face of a sectioned specimen (Fig. 4e), including transition zones between the stainless core and the mild steel casing. No intermetallic phase formation is detected. Through EBSD, close inspection of interface grain formation and crystallographic texture revealed strong ferritic orientation with {111} poles aligned with the cylinder surface tangent direction for the mild steel casing, with an observable concentration of {101} orientations in the immediate vicinity of the casing-core boundary (Fig. 6). Austenite formations were observed in the stainless core in the {101} orientation.

### Hardness and compression testing

Vickers hardness values and the testing location are shown in Fig. 4. Overall hardness for the bimetallic depositions averages close to 260 HV, with the stainless core averaging 249 HV and the mild steel casing averaging slightly higher at 277 HV. The average hardness for the top of the deposition was 241 HV, 246 HV for the middle, and 289 HV for the bottom, with a standard deviation of 36 HV, 26 HV, and 39 HV, respectively. This degree of variation is usual in DED-arc and is not unusually large. Similar variation is seen by other researchers conducting DED-arc studies[24,34]. Hardness values closer to the substrate are generally higher than those seen toward the top of the deposition. Hardness is not symmetric within the build plane, with values ranging from 394 HV to 234 HV when tested in the xy plane, perpendicular to the build direction (Fig. 4f).

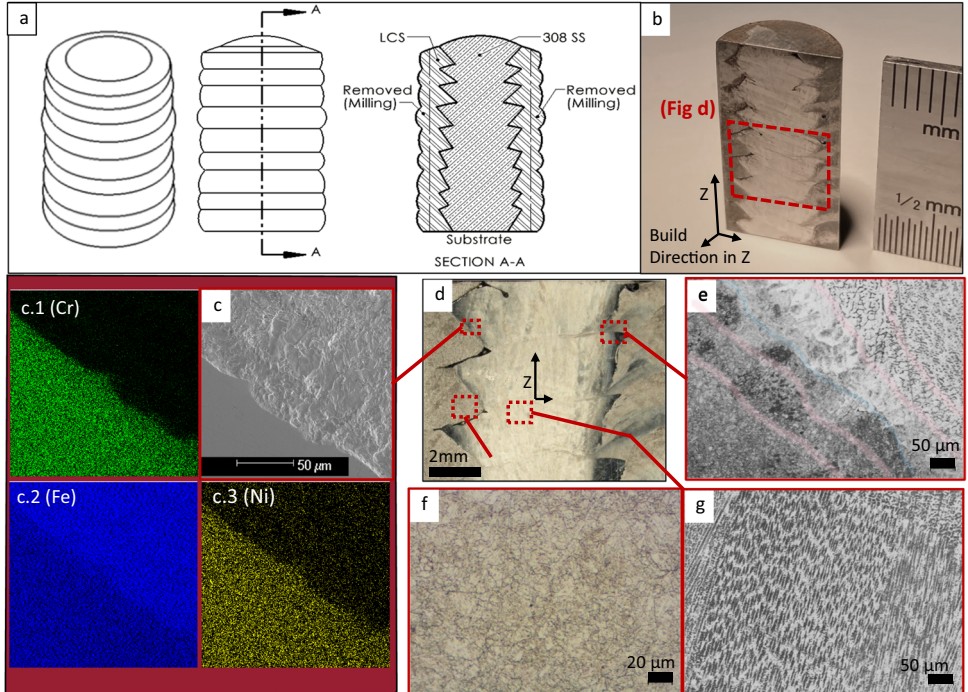

**Fig. 3 | Microstructural analysis and EDS. a** Schematic illustrating the as-deposited radial bimetallic structure arrangement, material removal boundaries for subsequent milling procedures, and the cut plane location used to provide microstructural analysis; **b** Polished cross-sectional image showing build direction and the interlocking zig-zag pattern of wedge-shaped protrusions of stainless material in the core into the mild steel casing, and vice versa. Also indicated is the location of subsequent micrographs; **c** SEM image of the weld interface between the 308L stainless core wedge and the corresponding portion of the mild steel casing; (c.1–c.3) EDS characterization insets indicating migration of Cr, Fe, and Ni; **d** Stereoscope image of sectioned bimetallic deposition, indicating locations for higher resolution micrographs and EDS; **e** microstructure observed at the upper interface between the stainless core and mild steel casing, with approximate boundaries between coarse reheated structure, transition zones, and fine structured new weld areas highlighted in red, and the approximate boundary between mild steel and stainless steel highlighted in blue; **f** equiaxed microstructure observed lower in the mild steel casing; **g** Columnar microstructure observed at the interlayer transition zone of the stainless core.

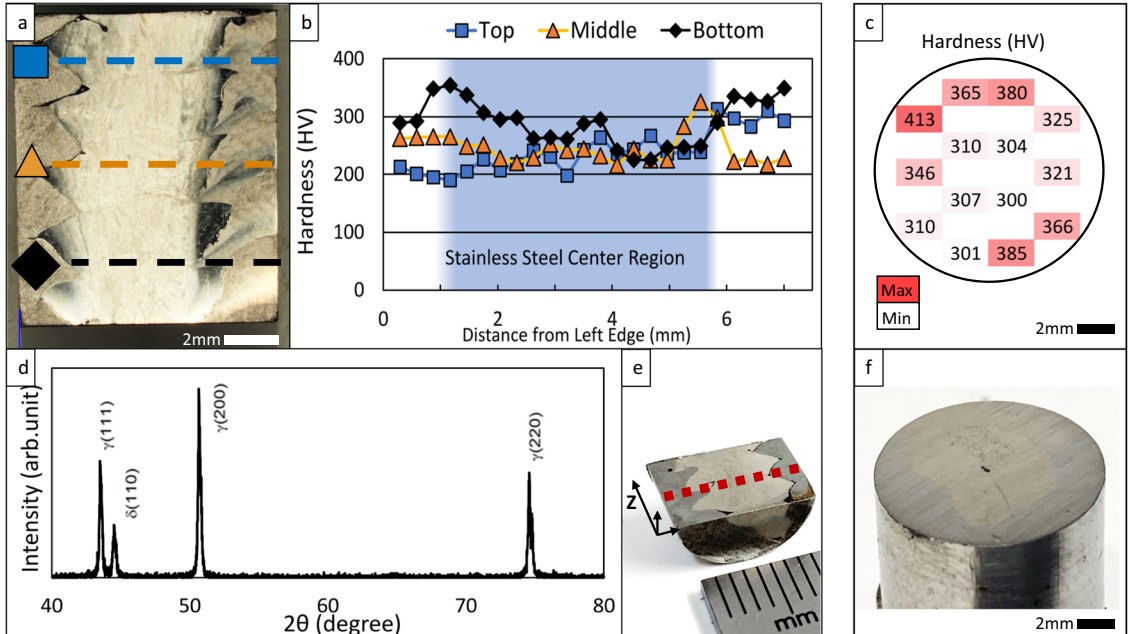

**Fig. 4 | Hardness and phase XRD analysis. a** Low magnification image of the polished and etched bimetallic specimen, with locations for hardness values indicated perpendicular to the build plane; **b** Hardness values collected horizontally across the face of the sectioned specimen; **c** Hardness values within a single xy build-plane; **d** Phase XRD results; **e** Diagram showing specimen orientation and pattern for phase XRD testing; **f** Image of the bimetallic specimen used in xy plane hardness testing, with bimetallic interface visible.

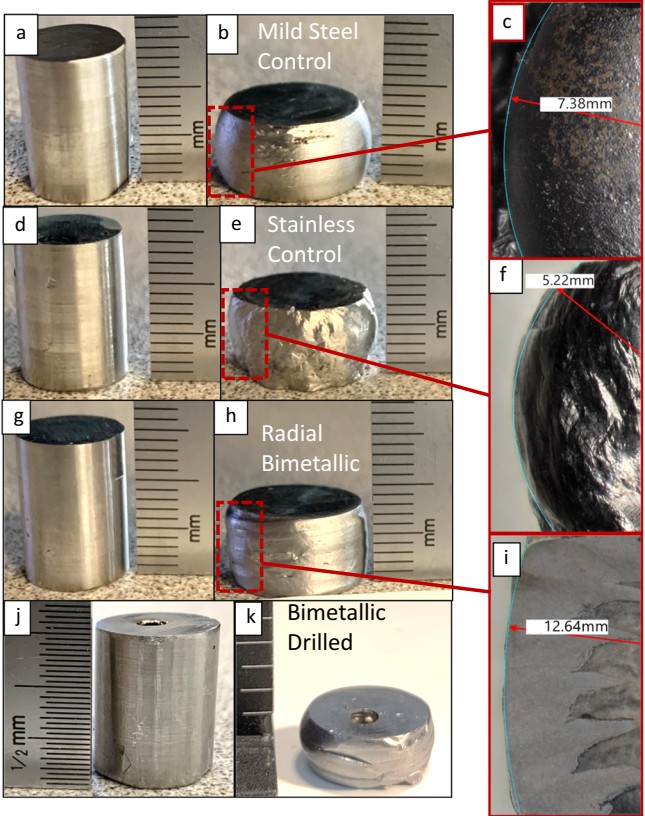

**Fig. 5 | Compression and deformation analysis. a** Mild steel monolithic compression specimen prior to the test; **b** Mild steel monolithic compression specimen after testing; **c** Radius of curvature for outward plastic deformation in mild steel specimen; **d** 308 L stainless monolithic compression specimen prior to the test; **e** 308 L stainless monolithic compression specimen after testing; **f** Radius of curvature for outward plastic deformation in 308 L stainless specimen; **g** Radial bimetallic compression specimen prior to the test; **h** Radial bimetallic compression specimen after testing; **i** Radius of curvature for outward plastic deformation in the radial bimetallic specimen; **j** Specimen with radial bimetallic variation and a drilled relief passage through the center of the stainless core; **k** Drilled radial bimetallic specimen after testing.

Compression samples for mild steel, stainless steel, and radial bimetallic specimens – before and after test evaluation – are shown in Fig. 5. Deformation from compressive loading was most significant in the stainless and mild steel control specimens but noticeably constrained in the bimetallic specimen (Fig. 5k). All sectioned specimens revealed gross defects at the interface between the core and the casing that remained as voids or cracks even after compression, although some regions had complete fusion at the interface between interlocking wedges of material (Fig. 6). Compression yield strength values obtained during uniaxial compression tests are presented in Fig. 7a. These compressive strengths averaged 370 MPa for mild steel controls, 346 MPa for stainless steel controls, and 493 MPa for radial bimetallic specimens. The compressive strength observed for the radial bimetallic specimens increased by 33% over mild steel controls and 42% over stainless steel controls. Compression testing was also conducted using a bimetallic specimen with a relief passage drilled directly through the center of the 308 L stainless core. The relief passage had a 3.175 mm diameter, sufficient to remove enough 308 L material to permit unconstrained plastic deformation during compressive loading concentric to the core. This specially prepared bimetallic specimen displayed a compression yield strength of 350 MPa and severe plastic deformation internally and externally when sectioned (Fig. 7f).

## Discussion

One of the main objectives of this work was to explore the potential advantages of multi-dimensional bimetallic DED-arc compositions and to introduce possibilities unique to AM structures with complex interfaces. To that end, radial bimetallic variation within a single horizontal plane was repeated continuously in the Z-axis build direction to perpetuate the interface throughout the entirety of the metallic specimen. Remarkably, the deposition and cooling of the bimetal couple occurred concentrically and nearly simultaneously. This is significant because the resulting interaction between the metals as they cooled together in the annular arrangement promotes hoop stresses completely incongruous with linear wall depositions, bimetallic or otherwise. This physical arrangement, coupled with the impact of increasingly significant DED-arc process parameters on material interactions, promotes the exploitation of bimetallic capability[23]. The bimetallic specimens thus produced were characterized by a complex interface that exhibited both metallurgical bonding and mechanical interlock, consistent up through the build layers from the base plate to the deposition crown. Mechanical testing and microstructural analysis confirmed the feasibility of producing radially variant bimetallic additive structures, with evidence for beneficial as-deposited structural residual stress due to a slight mismatch in CTE between the materials.

In typical bimetallic structures, intermetallic compounds are often formed along the interface between the component materials. These intermetallic compounds bear properties independent of the parent materials and metallurgically drive final part qualities detectable in compressive analysis or other mechanical testing. In these cases, the compressive strengths of bimetallic structures often register somewhere between, or slightly above, the parent material capabilities. It was expected that the bimetallic specimens in this current work would behave similarly. When analyzed, the as-deposited interface between deposited feedstocks showed slight migration of Cr and Ni elements from the stainless region into the mild steel casing (Fig. 3c). The contact between these two materials in the solidifying state would necessarily foster a gradient of some degree, and this was expected. At the same time, the minimal migration observed in the EDS data for these specimens prevents any conclusive observations on the occurrence of solid solution strengthening. Consistent with other research on DED-arc, columnar grain growth was observed independent of layer transitions[34,35]. Phase analysis conducted across the face of the sectioned bimetallic structure showed an unanticipated lack of new intermetallic phase formations (Fig. 4d)[36]. This may be attributable to similar thermal profiles experienced between the two base metals during deposition. When evaluated in compression, the bimetallic specimens exhibited compressive strengths on average almost 38% higher than the monolithic controls, with a standard deviation of only 17 MPa (Fig. 7a). Without significant quantities of intermetallic phase formations, this enhanced compressive strength downplayed any metallurgical bimetallic effect and promoted investigation of the mechanical interaction of the bimetallic couple.

The sectioned specimens show a distinct interface between the stainless core and the mild steel casing along, and relatively symmetric about, the Z-axis (Fig. 7b). This pattern forms a multi-dimensional, complex system of interlocking wedges between both materials, extending into and surrounded by each other (Fig. 7c). Interlocking, overlapping, or stratified interfaces have been seen and investigated, but in the linear form[35,37–39]. In the context of existing literature, the pattern presented in the current work is unique in its radial form and resultant physics, fundamentally departing from both planar interfaces and bimetallic walls. The concentric rings of material form cylindrically arranged bimetallic interfaces, with resultant hoop stresses unachievable by any currently available linear or stratified deposition methodology. The interlaced boundary between the core and the casing directly results from the deposition sequence and path

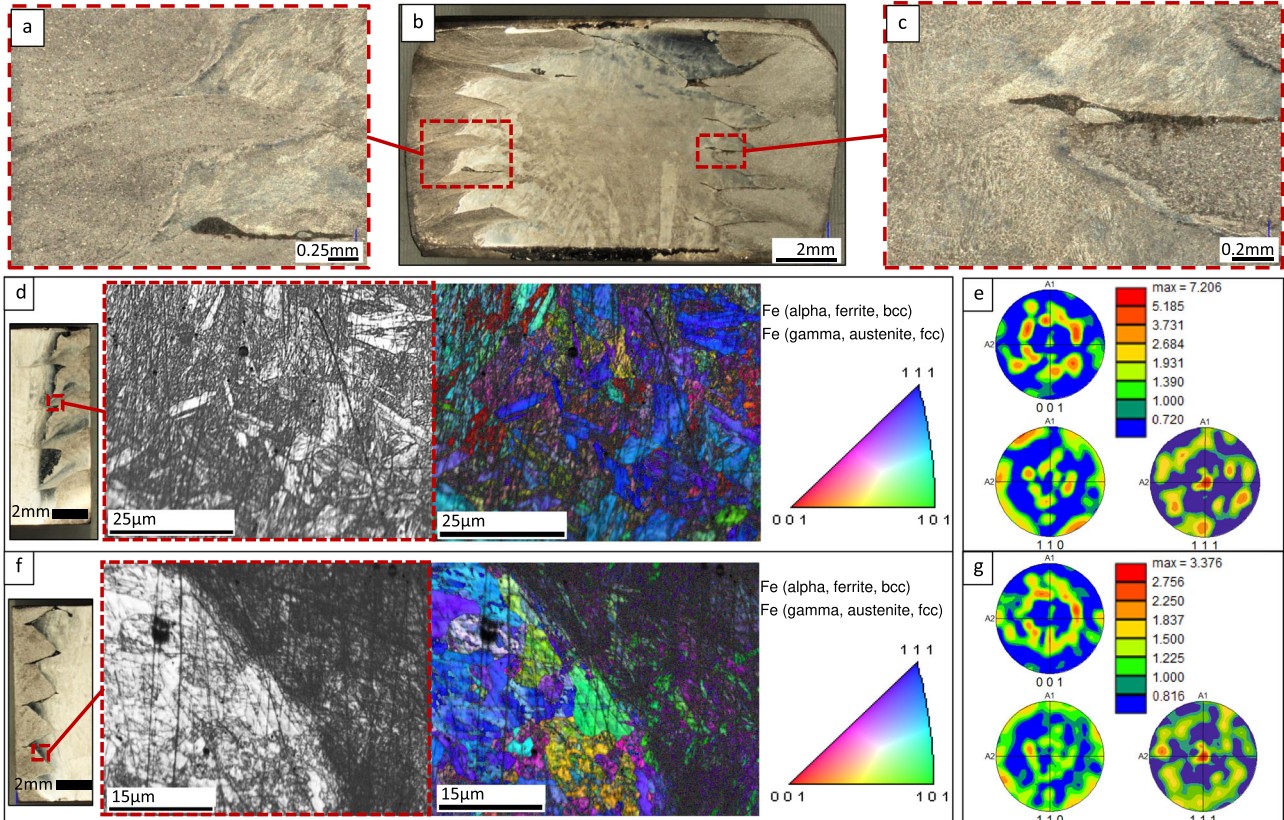

**Fig. 6 | Deformed bimetallic microstructure and EBSD analysis. a** Microstructure at the mild steel-stainless steel interface of a post-test bimetallic compression sample. A void is preserved after high compressive loading as the stainless wedge tip above it collapses downward; **b** Stereoscope image of the sectioned and compression-tested bimetallic specimen showing prevalent interface voids on the right side of the image correlating with a greater outward movement of the deformed mild steel on that side; **c** Microstructure and voids present at the mild steel-stainless steel interface in a post-test compression specimen; **d** Location of SEM image and EBSD orientation maps observed in the mild steel casing (color keys indicate poles aligned with the cylinder tangent directions); **e** {001}, {011}, and {111} pole figures for mild steel casing (normal direction is aligned with the cylinder tangent direction of the specimens); **f** Location of SEM image and EBSD orientation maps observed at the bimetallic interface between the core and casing; **g** {001}, {011}, and {111} pole figures for ferrite grains at the bimetallic interface.

planning. The casing was deposited first, following a circular path, and creating a circular bead surrounding a hollow depression. Before this bead could cool, and in an immediately following sequence, a separate weld system was activated along a separately controlled deposition path. This second deposition filled the hollow depression with core material, after which both depositions within the layer were allowed to cool simultaneously. It was noted that during core deposition, small overflow quantities of molten weld puddle regularly approached the crown of the encircling weld bead that would form the casing. The process repeats for each layer, with the circular unconstrained initial casing bead spreading at its base and narrowing at its crown – repeatedly forming the hollow depression into which the core material can flow. This continued sequencing forms the mechanical interlock observed in each sectioned bimetallic DED-arc sample and is sure to impact the mechanical properties and performance of the final part[25,40].

The thermal behavior of the two materials thus joined is necessary for the physical interaction and arrangement of the casing deposition encircling the core. The CTE for 308 L stainless is $9.6 \times 10^{-6}$ 1/°F. Mild steel has a smaller linear CTE of $6.5 \times 10^{-6}$ 1/°F. Due in part to this slight CTE difference, each layer of deposition adds net residual stress in the growing specimen, as indicated by predicted pressure concentrations at wedge vertices (Fig. 7d). Each additional disc of inner core deposition shrinks at a higher rate than the surrounding casing ring upon which it partially overlaps, creating a clamping relationship. Thus, constrained by the rapidly shrinking core, the outer casing adds further compressive hoop stress as it cools (Fig. 7e). This process repeats

throughout the deposition, creating a buildup of residual pressure between the interlocking wedges as molten weld puddles rapidly cool and reheat. This stress is assumed to remain as heating and cooling cycles are repeated until the completion of the build. The total effect imparted on the structure is similar to the function of cable tendons used to manufacture pre-tensioned concrete structures. In those applications, a cable is first tensioned, after which concrete is poured and cured around the cable. When the cable is released, it compresses the cured concrete casing. The structural enhancement and benefits of this technique are well documented. Similarly, residual compressive loads in the DED-arc bimetallic casing caused by the slight CTE mismatch and forced tensioning of the core will resist crack propagation and structural failure – although in the case of DED-arc, the tensioning occurs in situ layer by layer and simultaneously for both materials. To substantiate this concept, a simplified model (Fig. 7b) of the interlocking bimetallic structure was created to simulate pressure induced on the bimetallic interface by thermal contraction during final, post-deposition structure cooling. In a linear cool-down sequence from 500 °F to room temperature, pressure concentrations between the two material types at the vertices of the interlocks are predicted to peak at $1.8 \times 10^3$ MPa, but also appear consistently elevated along the underside of each stainless wedge. The mild steel casing also exhibited generally high-pressure intensity, between $1.4 \times 10^3$ MPa and $7.3 \times 10^2$ MPa in nearly all regions. This simulation approach was beneficial in validating the analytical results. To increase confidence in the accuracy of this simulation, a bimetallic specimen highly similar to the modeled specimen was heated to 550 °F and then cooled to room

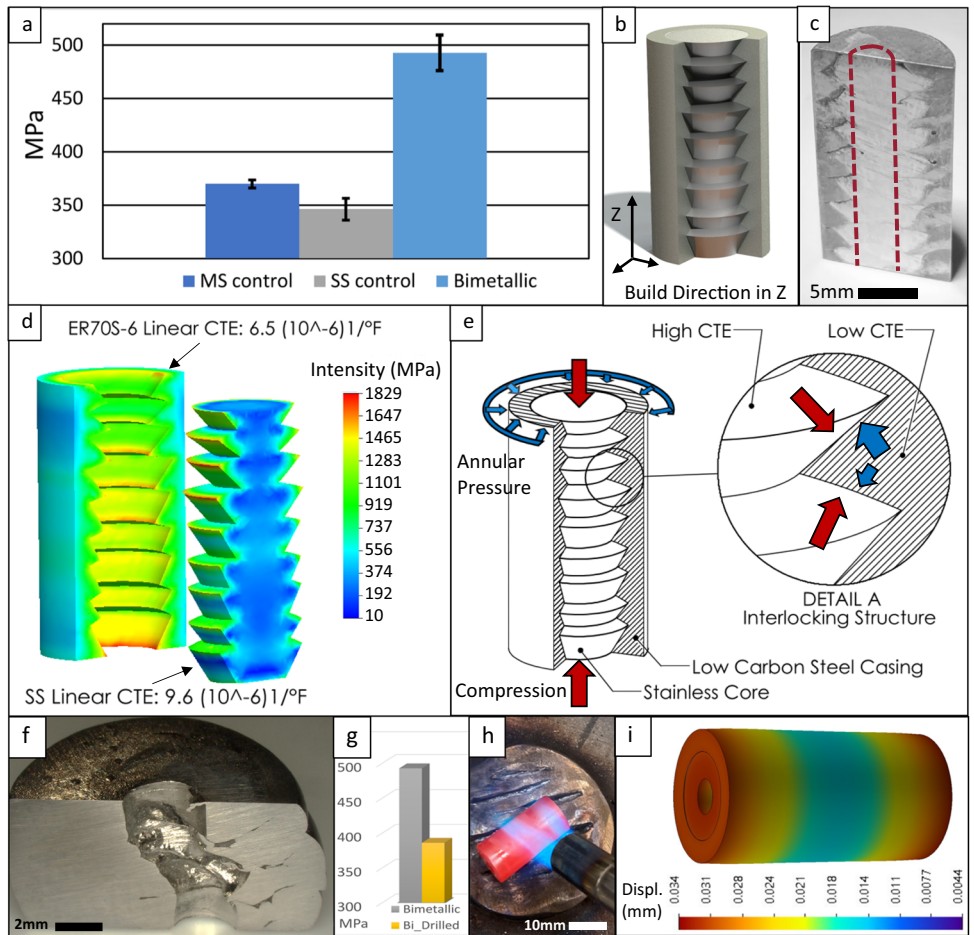

**Fig. 7 | Impact and role of residual pressure. a** Comparison of average compressive test results for monolithic and bimetallic specimens, with bimetallic compressive strengths averaging 37.8% above monolithic controls. Error bars indicate statistical uncertainty from repeated experiments; **b** 3D model of the multi-dimensional annular bimetallic cylinder, sectioned to reveal the interlocking interface between the 308 L stainless core and the mild steel casing; **c** Sectioned bimetallic specimen showing the as-deposited interface, as well as an illustration of the location for a drilled relief passage; **d** Analytical model indicating mechanical interactions between the two materials with pressure exerted at wedge boundaries; **e** Illustration exploring the effect of slight CTE differences, depicting each additional disc of inner core deposition shrinking at a higher rate than the surrounding casing ring upon which it partially overhangs, creating a clamping relationship and net residual stress in the growing specimen. The outer casing adds further compressive hoop stress as it also cools, constrained by the rapidly shrinking core; **f** Post-compression bimetallic specimen with a passage drilled through the core, showing severe deformation; **g** Compression test results for the unconstrained drilled bimetallic specimen compared to the as-deposited bimetallic specimen; **h** Heating of a bimetallic specimen to 550 °F, for subsequent cooling to room temperature to validate the accuracy of the analytical model; **i** Analytical model predicting the percent change in overall physical displacement during cool-down from 550 °F to room temperature.

temperature (Fig. 7h). The percent change in overall physical displacement between the elevated and cooled temperature was calculated and compared to the percent change in overall physical displacement predicted by the model (Fig. 7i). This calculation revealed a 14.8% difference between the actual and predicted displacements, which does not provide evidence against the modeling technique. Taken as a whole, the analytical model indicates residual stress from simultaneous cooling that induces compression in the mild steel casing, attributable to thermal contraction of the core at a slightly different rate than the casing. The suggested comparability to a tensioned tendon in prestressed concrete is not rejected, and the general effect, like prestressed concrete, is a strengthening relationship when mechanical properties are evaluated.

During compression testing of the bimetallic specimens produced for this study, it is supposed that residual compressive stress in the mild steel casing resists crack propagation. EBSD characterization of the crystalline texture at the interface found a surprising concentration of {111} fiber texture ferrite grain orientations aligned with the cylinder tangent direction in the mild steel, with traces of {101} orientations in the stainless steel. Very few {101} oriented grains were observed in the casing, which would have been anticipated due to alignment with the build direction. The effect of grain morphology on solidification residual stresses is of interest but considered beyond the scope of this article. Further details related to DED-arc can be found in[41-43].

The observed pre-tensioned behavior also seems to constrain plastic deformation. Severe deformation is a typical result of compression testing and was expected and observed in the monolithic control specimens. Each monolithic specimen exhibited a large outward deformation of the material, observable as specimen convex side curvature in Fig. 5 for loads averaging 358 MPa. The compression-tested bimetallic specimens, however, had much less deformation despite sustaining higher loads that averaged 493 MPa. Sectioned and polished post-test bimetallic compression specimens showed comparatively little displacement of material, with interlocking relationships preserved (Fig. 6). The displaced material was not symmetrical, however, with a radius of curvature reflecting a 22.3% decrease between the side corresponding to full fusion between wedges and the

side characterized by voids or incomplete fusion. From these results, it is hypothesized that the outward motion of casing material is effectively prevented as the geometrically constrained core clamps down on the interspersed wedges of mild steel. When unrestrained by poor fusion in the bimetallic arrangement, such movement is unconstrained and tends toward mechanical performance that is more characteristic of a monolithic structure.

To further substantiate this relationship between residual hoop stress and resistance to plastic deformation during loading in the concentric bimetallic couple, a hole was drilled through the center of the 308 L stainless core. The anticipated effect of a relief passage so arranged was to back-track the strength enhancements attributed to the sequence of a rapid annular combination of concentric bimetallic material and the residual hoop stresses generated during the simultaneous cool-down of the metals. This relief passage would reduce residual pressure between the core and the casing by allowing internal plastic movement of material during compression testing. Compression testing was completed under conditions otherwise identical to non-drilled bimetallic specimens. Similar to the monolithic specimens, deformation was no longer constrained, and the outward plastic movement of the mild steel casing was severe (Fig. 5k). The specimen reached failure quickly, with an apparent collapse of the stainless core into the drilled passage (Fig. 7f). All strengthening behavior of the bimetallic relationship disappeared, with the bimetallic specimen exhibiting a compressive strength within 7 MPa of the 358 MPa average for non-bimetallic monolithic controls (Fig. 7g). The relief passage allowed material flow during loading, thereby providing dissipation of the residual clamping pressure and hoop stress generated during deposition cooling.

The pre-tensioned concept for bimetallic AM has significant ramifications in additive design strategies for the future. The ability to create strengthening relationships layer by layer through radial variation will undoubtedly capture attention for various applications. Shrink fits come immediately to mind, where high radial pressures between dissimilar components are relied upon to reduce relative motion or transmit torques. Simultaneous fabrication of crankshafts, railway wheels, bands, turbine discs, and machine tooling are additional potential applications. Large-scale spacecraft may include AM-produced structural members capable of spanning large distances with a central pre-tensioned material core that compresses the entire structure. With such applications in mind, there are challenges that further study must address. For instance, the physical arrangement of two dissimilar materials and their interaction in the final build suggests the importance of deposition path planning and sequencing. Although that was not the focus of this study, it will impact final characteristics. Sharp changes in material composition and stress concentrations at the tips of the interlocking wedges may also promote failure in high-cycle fatigue service situations, and dissimilar bimetallic couples are proven susceptible to galvanic corrosion[44]. Another concern related directly to the findings of this study is the variability observed in hardness throughout the deposition (Fig. 4b, c). Hardness values were expected to be greatest at the base of the deposition, with accelerated cooling due to proximity to the substrate chill plate[45]. This relationship was confirmed, with base hardness averaging 289 HV and final layer hardness averaging 241 HV. However, the non-symmetric distribution of general-area hardness values within a single xy plane deposition layer was not anticipated (Fig. 4c). The momentary pause in torch movement during arc-start and arc-end causes additional thermal input at these locations during deposition, in a slightly delayed two-step fashion. Slower cooling times necessarily experienced at these deposition positions would translate into a softer microstructure, thus explaining the hardness observed. The variation observed within one of these layers is, therefore, likely to be a direct outgrowth of deposition path planning, further supporting path planning as a factor of interest for future work.

Specimens produced in this study provide a parametric basis for additional research in various advanced materials, where bimetallic arrangements with complex interface locations bear the potential for significantly augmented performance properties. One such research direction is illustrated by the bimetallic specimen fabricated with a drilled passage through the center of the stainless core (Fig. 2d, f). Conceived as a way of verifying deformation constraint as a driver of compressive strength, this specimen also represents a short section of corrosion-resistant pipework. The stainless core is a corrosion-resistant conduit, while the mild steel casing becomes a stiffening and cost-saving element. This feasibility concept is easily extrapolated to more advanced materials and service applications where the combination of characteristics between dissimilar materials in a single component exceeds either material's capability.

This work used dual welding torches to deposit dissimilar metal wire feedstocks together in the same plane with a radial variation using wire arc additive manufacturing. The constituent materials were also deposited to form individual monolithic cylinders for mechanical performance comparison. The radial bimetallic pattern was repeated consistently throughout subsequent layers in the Z-axis build direction. Thus, multi-dimensional cylindrical specimens were characterized by a complex interlocking bimetallic interface between a stainless-steel core and a mild steel casing. The interlocking relationship between the core and the casing was discovered when sectioned and polished, with microstructural grain growth transcending interface boundaries. EDS showed Cr and Ni migration from the stainless into the mild steel, but XRD phase analysis detected no new formations. Compression testing revealed a 33% relative increase in strength for radial bimetallic specimens over monolithic mild steel and a 42% relative increase in strength over monolithic stainless steel. Thermal stress analysis of the CTE mismatch between the two materials during the final cool-down suggested a residual stress effect that resists crack propagation. A relief passage drilled through the core of a bimetallic specimen allowed unconstrained plastic deformation and relieved any residual stress concentrations. The resulting compression strength was reduced by 28% from comparable non-drilled bimetallic specimens, confirming the concept. As-deposited hardness in the mild steel casing was higher than in the stainless core, with vertical variation consistent with thermal dissipation attributes but unexpectedly varied within a single deposition layer. The findings in this study encourage the investigation of radial bimetallic arrangements and pre-tensioned AM structures. Most significantly, a basis is provided for more complex arrangements and functionalities in bimetallic components produced by DED-arc.

## Methods

### Processing of radial bimetallic structures via DED-arc

308 L stainless steel control samples were deposited using a Titanium Unlimited 200 DC welder, shielded by a mixture of 80% Ar, and 20% $CO_2$ at 18 L/m. The wire arc was struck using 19 V and 230 in/min wire feed. A mild steel substrate was used, and twelve layers were deposited with a final as-deposited height for each specimen of 15 mm, with a 10 mm diameter. ER70S-6 mild steel control samples were also deposited using a similar Titanium Unlimited 200 DC welder, shielded by a mixture of 80% Ar and 20% $CO_2$ at 18 L/m. The wire arc was struck using 15.1 V and 168 in/min wire feed. For bimetallic specimens, ER70S-6 mild steel was first deposited using the same equipment for the control samples but with a voltage setting of 15.1 V. Deposition parameters for 308 L stainless steel in the bimetallic specimens were generally the same as for the individual control specimen but with a voltage of 21 V and a 349 in/min wire speed. These voltage and wire speed settings promoted bead height deposition commonality between the two different feedstocks for each layer. Concentric deposition of the ER70S-6 mild steel casing in the bimetallic specimen was also accomplished consistent with the parameters used for the

**Table 1 | Chemical composition of wires used to produce monolithic and radial bimetallic specimens[46]**

|         | %C   | %Mn  | %P    | %Si | %Cu  | %S    | %Cr  | %Ni  | %Mo | %Fe  |
|---------|------|------|-------|-----|------|-------|------|------|-----|------|
| ER70S-6 | 0.08 | 1.52 | 0.009 | 0.8 | 0.2  | 0.012 | -    | -    | -   | Bal. |
| ER3O8L  | 0.02 | 1.7  | 0.002 | 0.4 | 0.21 | -     | 20.5 | 10.5 | 0.3 | Bal. |

mild steel control structures. The final as-deposited structure height for bimetallic specimens was 18 mm high, with diameters of 12 mm after 12 deposition layers. The chemical compositions of these materials are given in Table 1.

The overall process is shown in Fig. 2a. Concentric deposition paths were programmed using DXF files and PathPilot® CNC controller conversational programming. These deposition paths were independently executed in rapid succession using a Tormach® 770 M CNC mill. The torch travel speed for 308 L stainless steel control samples was 330 mm/min, and 250 mm/min for ER70S-6. Torch travel speed for bimetallic specimens was controlled at 250 mm/min for both material types, although SS core deposition required minimal torch movement. All samples were deposited on mild steel substrates positioned on top of a substrate chill plate (Fig. 2b). As-deposited specimens were first machined on a Tormach® 1100MX CNC mill and then removed from their substrate using a Jet horizontal bandsaw. The final diameter for all cylindrical bimetallic and control specimens was 10 mm, and all specimens had a 15 mm final height.

## Microstructural and phase analyses

For microstructure characterization, samples were cut in half along their long axis (Fig. 3a) to expose the bimetallic relationship using a low-speed diamond saw lubricated and cooled with mineral oil. Wet-grinding progressed sequentially with 80-1200 grit SiC sandpaper. Polishing was followed sequentially with 0.5-0.05 μm alumina polishing powder suspended in DI water for each sequence. Sectioned specimens were ultrasonically cleaned for 15 min. in 50% ethanol. Samples were etched for 20 s in a solution of 10 mL HNO$_3$, 10 mL acetic acid, 15 mL HCl, and four drops of glycerol. Optical microscope images were captured with an Axiocam 105 and Axiocam ERC5S. The energy dispersive spectroscopic (EDS, EDAX by Ametek, PA) analysis was done using a field emission scanning electron microscope (FESEM, FEI-SIRION, Portland, OR). For phase analysis, X-ray diffraction (XRD) was done using a Cu k-alpha radiation (1.54 angstroms at 40 kV and 20 mA) on a Rigaku mini flex 600 X-ray Diffractometer equipped with a 2-D General Area Diffraction Detector (GADDS) mounted on a theta-theta goniometer. The scanning speed was 5° per min between the 0 to 100 degrees range of 2θ. Raw data were processed with Rigaku PDXL software. Electron backscatter diffraction (EBSD) was used to characterize crystallite orientations with a scanning electron microscope (Apreo 2 FEG-SEM) and EDAX Velocity system. An accelerating voltage of 20 keV, stage tilt of 70 degrees, and 0.10-micron step size at a working distance of 8.6 mm was used to acquire the Kikuchi patterns. TSL OIM software was used to provide analysis of the EBSD data, discarding data with a confidence index of less than 0.1. The EBSD data were not manipulated other than discarding the low confidence points after a Confidence Index standardization routine.

## Compression and hardness testing

Compression data was gathered through quasi-static testing with a UTM-HYD Instron servo-hydraulic machine (600DXS, Grove City, Pennsylvania). A constant crosshead displacement rate of 1 mm/min was used until a maximum load of 20,000 kg or complete failure was achieved. In all cases, the specimen height-to-diameter ratio was 1.5, following ASTM E9, Standard Test Methods of Compression Testing of Metallic Materials at Room Temperature. Compression testing was chosen primarily due to the cylindrical nature of the radially arranged

bimetallic specimen and for convenience in sample preparation. Compressive loading was the most appropriate mechanical evaluation for the role the casing would play in restraining deformation throughout the structure. Hardness testing was performed using a Phase II Plus, Micro Vickers hardness tester (Upper Saddle River, NJ, USA). Hardness indents were made on the sample cross-section in three rows, across the top, middle, and base, perpendicular to the Z build direction (Fig. 4a). Indents had a 0.098 N load, with 15 s dwell. Specimens were ground and polished progressively using SiC sandpaper from 80 to 1600 grit size and polished with alumina media. Hardness values were also collected using identical settings for the drilled bimetallic cylinder in the xy plane, centering around the drilled passage (Fig. 4c, f).

## Analytical analysis

A representative bimetallic specimen was modeled with mild steel and 308 L stainless material properties assigned to the respective portions in the fully built and machined condition. The post-deposition cool-down sequence was modeled using SolidWorks Simulation 2020 SP3. Bimetallic interfaces were fixed, while all other structure was unconstrained. Specimen cooling was simulated linearly from 500 °F to 69 °F ambient room temperature, with a 308 L stainless coefficient of thermal expansion (CTE) of $9.6 \times 10^{-6}$1/°F and the CTE for mild steel set at $6.5 \times 10^{-6}$1/°F. The pressure generated between the core and casing during cooling was evaluated. Simulated linear expansion and contraction values were compared to actual values obtained by subjecting an identical physical specimen to identical cooling conditions. Percent change in overall physical displacement was calculated to provide confidence in the accuracy of the simulation steps and boundary conditions.

## Data availability

The authors declare that all data supporting the findings of this study are available within the paper.

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

## Acknowledgements

The authors would like to acknowledge financial support from the National Science Foundation under the grant CMMI 1934230 (PI – A.B.). The authors also acknowledge experimental help from Prof. David Field and Ms. Claire Adams for EBSD imaging, and Mr. Aruntapan Dash for X-ray diffraction studies.

## Author contributions

L.S. designed and executed the experiments, analyzed the data, conceived the concept, and wrote the manuscript with assistance from all authors. E.R. carried out experiments and analyzed data. A.B. contributed to project conceptualization, supervision, funding acquisition, experimental design, data analysis, reviewing, and manuscript editing.

## Competing interests

The authors declare no competing interests.
