## [Peer Review File · Nature Communications]

Radial Bimetallic Structures via Wire Arc Additive ManufacturingREVIEWER COMMENTS

Reviewer #1 (Remarks to the Author):

Comments:

In this present work, the authors did an interesting and systematic work to investigate the radial bimetallic structures following biomimetic design concepts through WAAM. It is real a good topic and a good extension of the WAAM technology. Meanwhile, the structure of this manuscript is well organized, experimental procedure is reasonable and the conclusion is convincing. Therefore, I guess the readers can get a good idea of the context and the issues.

However, several issues should be addressed:

1. On Page13, the authors mentioned that: “All sectioned specimens revealed gross defects at the interface between the core and the casing that remained as voids or cracks even after compression, although some regions had complete fusion at the interface between interlocking wedges of material (Fig. 5I & L)”. However, carefully check the Fig. 5I & L, it is really hard to find the complete fusion, especially from the Fig. 5L, which is only a macro topography. Please give more explanation.
2. Please pay more and more attention on the details, such as the use of the punctuation. On Page14, line3: “...wedges during compression. (D-G) Compression...”. It should be “...wedges during compression; (D-G) Compression...”. Please check this detail carefully.
3. On page14, line4, “(D-E)” should be changed to “(D, E)”, right? Check the (F, G)
4. On Page17, line4, the full stop before “(G)” should be changed to a semicolon. “... deformation of both the core and the casing. (G) Compression ...” should be “... deformation of both the core and the casing; (G) Compression ...”.
5. On page17, line4, for (G), the authors point out that the compression testing of the drilled bimetallic specimen is similar to the non-bimetallic specimens. However, the results of the pure metals samples did not list in this present work. Although the authors mentioned that the testing had previously, there were no dates.
6. In this work, it can be found the zig-zag pattern on wedge-shaped protrusions of stainless materials in the core into the mild steels casing. Meanwhile, the pure metals (stainless and mild steels) also fabricated in this work. My question is that: the zig-zag pattern can be found in the pure metal by the WAAM?
A very important set of drilling experiment was proposed in this work, which support the results of the increase in compressive strength caused by residual stress has disappeared. It is pity that the drilling experiments are not convincing enough. How to choose the size and position of the hole. More

illustration should be added.

As can be seen in Figure 4F, only a hole was punched at the junction of the two metals. On Page 19, the authors pointed out that "However, the non-symmetric distribution of hardness values within a single X-Y plane deposition layer was not anticipated (Fig. 4C). A possible explanation for the conditions thus observed is the adjacent locations of arc-strike and arc-end with their attendant pause times (Fig. 2A)." I think the above statement is not rigorous. Three different positions holes should be discussed.

7. Even if no new compounds are generated, the solid solution strengthening caused by element migration between bimetallic samples should also be taken into account, However, there is no quantitative comparison between metallurgical factors and mechanical factors.

8. In Fig 4, why do the hardness values in the inner region of the bottom fluctuate greatly? According to fig. 3(G), the microstructure in the middle is very uniform. Please explain it.

9. From Fig 3(B), non-fusion and porosity can still be observed clearly. The parameters of WAAM is not mature enough, right? How to obtain the good metal structures without these faults?

10. The modeling technique is presented in this work. However, it is best to illustrate more details on this simulation. How to build the model, how to set he boundary conditions,.... It is very useful to the readers.

11. For the bimetals structural by the WAAM, why choose the compression test, not the tensile test? An explanation of why the authors did these various experiments should be provided.

There is continued interest in this manuscript. I would be very glad to re-review the paper in greater depth once it has been edited because the subject is interesting.

Reviewer #2 (Remarks to the Author):

In this work, the authors investigated the fabrication of radial bimetallic structures via wire arc additive manufacturing. The idea behind the research is very interesting and the manuscript is well written and presented. The work can be accepted for publication after the following revisions are made:

1. The introduction is somewhat long. I would recommend to be shortened and more focused on the topic.

2. As the authors stated, AM provides design flexibility. The dissimilar metal in the shape of thin-wall cylinder geometry has been fabricated using centrifugal casting before. It would be beneficial to compare these two methods and discuss the advantages of AM over the more conventional casting method.

3. The captions of images are long. Rather than describing the image, they seem to discuss the image. I

would recommend shortening them and focusing on the description of the image.

4. Simulation steps need to be explained further in the material and method section, or in a Supplementary Information document.

5. A colormap scale bar should be added to Fig. 6I.

6. Finally, the readers may benefit from further explanations regarding the effect of residual stresses on distortion and mechanical properties.

A great extension of this work could be examining the structure under more realistic loading on the bones, such cyclic loading. I would recommend the authors to consider this in their future research.

Reviewer #3 (Remarks to the Author):

The work is centered around production of a radial interface (i.e., bimetallic wall) in contrast to a planar interface resulting from a vertical/horizontal deposition of layers in the WAAM process. The technique presents an original approach from that aspect, and in fact, different WAAM techniques that produce bimetallic components have been reported previously which resulted in improving the overall mechanical performance (e.g., refs. [1,2]). While the authors' technique (radial bimetallic structure) presents a meaningful advance to the field (evident by the improved strength from the compression tests), I fail to see why it is of much significance than those other techniques already published in [1,2] which also provide additional strengths, albeit in other testing regimes. The radial bimetallic interface in this paper is indeed complex, however, there is no comparison against its planar counterparts to justify its significance in that sense. I have the following concerns/questions which I hope the authors can address:

1. I am unable to support the authors' claim regarding the 'pre-tensioned concept' without in-depth microstructural analyses and discussion on and around the bimetallic interface (certainly EBSD and possibly TEM). Here, the paper can investigate the interface before and after compression to analyze the role of microstructure (crystallographic texture/grain morphology, etc.) on the attained mechanical strength. The EDS/EDX data which the authors provide can be expanded here also.

2. The drilled bimetallic specimen exhibited a strength of 350 MPa, which is less than the ER70S-6 (370 MPa) and very close to the SS specimen (346 MPa) (page 13). Including a comparison with drilled ER70S-6 and drilled SS seems logical to me for a fair comparison.

3. While drilling the bimetallic specimen does relieve stress, it also provides more means for material to flow, hence, expectedly reduces strength. Please discuss this.

4. It is essential to include the chemical composition of the two alloys in a stand-alone table.

5. Discuss the zig-zag feature of the bimetallic wall in context of existing literature (e.g., [3] and a very recently published paper [4], of course these references are not exhaustive)

Other minor comments:

1. Introduction is too broad. I suggest gearing it a little bit for a slightly knowledgeable target audience to shorten it.
2. Panels E, G, M in Fig. 5 should be the star of the show, because they immediately show the benefit. Hence, I recommend making them the highlight of the figure.
3. Kindly be consistent with the terminology/acronyms (e.g., low carbon control vs. mild steel, 308SS vs. 308L).
4. Remove unnecessary repetition of the experiments or motivation (e.g., beginnings of chapters 3.0 and 4.0).
5. Use sequential referencing to figures and figure panels (i.e., Fig. 1a, Fig. 1b...etc. not Fig. 1a, Fig 4b, Fig. 1b, Fig. 2c), thereby making it easier for the reader to follow without flipping pages/scrolling many times).

Response to reviewers' comments

Reviewer #1

General comments

In this present work, the authors did an interesting and systematic work to investigate the radial bimetallic structures following biomimetic design concepts through WAAM. It is really a good topic and a good extension of the WAAM technology. Meanwhile, the structure of this manuscript is well organized, experimental procedure is reasonable and the conclusion is convincing. Therefore, I guess the readers can get a good idea of the context and the issues.

Response: We appreciate this positive feedback. We have addressed all comments raised by the reviewer, and a detailed response is given below.

Comment 1. On Page13, the authors mentioned that: "All sectioned specimens revealed gross defects at the interface between the core and the casing that remained as voids or cracks even after compression, although some regions had complete fusion at the interface between interlocking wedges of material (Fig. 5I & L)". However, carefully check the Fig. 5I & L, it is really hard to find the complete fusion, especially from the Fig. 5L, which is only a macro topography. Please give more explanation.

Response: The incorrect figure reference has been changed to correctly indicate Fig. 5A & C. Figure and caption subsequently edited per **Reviewer #2 - comment 3**.

Comment 2. Please pay more and more attention on the details, such as the use of the punctuation. On Page14, line3: "...wedges during compression. (D-G) Compression...". It should be "...wedges during compression; (D-G) Compression...". Please check this detail carefully.

Response: Done. Figure caption punctuation has been corrected. Punctuation has been carefully reviewed throughout the entire text.

Comment 3. On Page14, line4, "(D-E)" should be changed to "(D, E)", right? Check the (F, G)

Response: Done. The figure caption has been corrected for uniformity.

Comment 4. On Page17, line4, the full stop before "(G)" should be changed to a semicolon. "... deformation of both the core and the casing. (G) Compression ..." should be "... deformation of both the core and the casing; (G) Compression ...".

Response: Done. The figure caption has been corrected for uniformity in punctuation.

Comment 5. On page17, line4, for (G), the authors point out that the compression testing of the drilled bimetallic specimen is similar to the non-bimetallic specimens. However, the results of the pure metals samples did not list in this present work. Although the authors mentioned that the testing had previously, there were no dates.

Response: References to previous testing of pure metals have been edited for clarity. Compression test results for the pure metal samples are presented in section 3.2, “Hardness and compression testing,” paragraph two.

Comment 6a. In this work, it can be found the zig-zag pattern on wedge-shaped protrusions of stainless materials in the core into the mild steels casing. Meanwhile, the pure metals (stainless and mild steels) also fabricated in this work. My question is that: the zig-zag pattern can be found in the pure metal by the WAAM?

Response: It is logical to suppose that both bimetallic and monolithic specimens formed by WAAM would show similar WAAM-related attributes. However, the zig-zag pattern detailed in this work is directly attributable to the use of two different alloys and is the signature of the interface. In a pure metal, there will be no bimetallic interface.

Comment 6b. A very important set of drilling experiment was proposed in this work, which support the results of the increase in compressive strength caused by residual stress has disappeared. It is pity that the drilling experiments are not convincing enough. How to choose the size and position of the hole. More illustration should be added.

Response: Done. The drilled relief passage placement and size details have been added to section 4.0. An illustration of the hole location has been added to Figure 6. Additional rationale for the drilling experiment has been added to section 4.0, with a new paragraph dedicated to this topic.

Comment 6c. As can be seen in Figure 4F, only a hole was punched at the junction of the two metals. On Page 19, the authors pointed out that “However, the non-symmetric distribution of hardness values within a single X-Y plane deposition layer was not anticipated (Fig. 4C). A possible explanation for the conditions thus observed is the adjacent locations of arc-strike and arc-end with their attendant pause times (Fig. 2A).” I think the above statement is not rigorous. Three different positions holes should be discussed.

Response: The quoted statement has been clarified in its discussion of single-plane hardness variation. It is important to note that the discussion concerns general-area hardness values observed due to deposition path planning rather than hardness, specifically at the bimetallic interface, which is discussed elsewhere in the manuscript. Additionally, the function of the drilled passage was to provide relief for potential residual stress by enabling plastic movement in the core (see response and edits made for comment 6b).

Additional hardness testing of a new bimetallic specimen was performed to support this response further, as shown in the figure immediately below **Fig 1**. The bimetallic core boundary is evident upon close viewing of **Figure 1A**. After a baseline hardness map (HV) was generated on the face of this specimen, three holes were drilled at various locations respective to the core (**Fig1B**). Hardness (HV) was then again measured in approximately the same positions as permitted by the absence of material. No significant impact on general-area hardness was observed due to drilling activity, regardless of the relief passage location near the specimen core.

To avoid confusion for the reader, Figure 4 in the manuscript has been modified with images showing hardness in the XY plane that does not simultaneously introduce the concept of the relief passage, which is more appropriately and fully addressed elsewhere in the manuscript.

Figure 1: Additional bimetallic cylinder for hardness testing and investigation on the role of hole location in determining general-area hardness values in the X-Y plane

Comment 7. Even if no new compounds are generated, the solid solution strengthening caused by element migration between bimetallic samples should also be taken into account, However, there is no quantitative comparison between metallurgical factors and mechanical factors.

Response: This is a very good point. For other bimetallic compositions, solid solution strengthening and migration of alloying elements may pose challenges that influence mechanical properties. This is often seen. In this particular bimetallic stainless and mild steel system, the concern would be the migration of chromium and nickel. However, the EDS analysis (**Fig 2 immediately following**) showed very little evidence of migration, as shown in the figures below. Based on those results, we are reluctant to speculate that solid solution strengthening is occurring. The manuscript has been modified to clarify this position.

Figure 2: EDS results of element migration observed at the bimetallic boundary

Comment 8. In Fig 4, why do the hardness values in the inner region of the bottom fluctuate greatly? According to fig. 3(G), the microstructure in the middle is very uniform. Please explain it.

Response: The average hardness for the top of the deposition was 241 HV, 246 HV for the middle, and 289 HV for the bottom, with a standard deviation of 36 HV, 26 HV, and 39 HV, respectively. This degree of variation is expected in WAAM and is not unusually large. Similar variation is seen by other researchers conducting WAAM studies [24], [35]. This explanation has been added to the text.

Comment 9. From Fig 3(B), non-fusion and porosity can still be observed clearly. The parameters of WAAM is not mature enough, right? How to obtain the good metal structures without these faults?

Response: Absolutely. As WAAM matures generally as a technology, parameters for quality depositions will undoubtedly continue to evolve. The authors are unaware of any radial bimetallic WAAM elsewhere in the literature, and by introducing radial bimetallic WAAM as a new specialty subset of the art, inevitably, faultless depositions through improved path planning and welding parameter manipulation will soon follow.

Comment 10. The modeling technique is presented in this work. However, it is best to illustrate more details on this simulation. How to build the model, how to set the boundary conditions,.... It is very useful to the readers.

Response: Done. Details on modeling methodology and boundary conditions have been added.

Comment 11. For the bimetal structural by the WAAM, why choose the compression test, not the tensile test? An explanation of why the authors did these various experiments should be provided.

Response: Done. The explanation for the use of compression testing has been added. Briefly, the compression test is an effective measure of the role of the stainless steel casing in restraining deformation throughout the structure. Fabrication of tensile coupons would also have prohibitively difficult.

General Comment. There is continued interest in this manuscript. I would be very glad to re-review the paper in greater depth once it has been edited because the subject is interesting.

Response: The interest is appreciated, and comments are found very constructive.

Reviewer #2

General comments

In this work, the authors investigated the fabrication of radial bimetallic structures via wire arc additive manufacturing. The idea behind the research is very interesting and the manuscript is well written and presented. The work can be accepted for publication after the following revisions are made:

Response: We appreciate this positive feedback. We have addressed all comments raised by the reviewer, and a detailed response is given below.

Comment 1. The introduction is somewhat long. I would recommend to be shortened and more focused on the topic.

Response: The introduction has been shortened, and we have tried to edit it with a greater focus on the subject matter.

Comment 2. As the authors stated, AM provides design flexibility. The dissimilar metal in the shape of thin-wall cylinder geometry has been fabricated using centrifugal casting before. It would be beneficial to compare these two methods and discuss the advantages of AM over the more conventional casting method.

Response: This comparison has been added to the introduction. Conventional casting methods, such as centrifugal casting, can produce various complex shapes. However, AM is fundamentally a tool-free manufacturing approach, where multiple design changes or variations can be easily incorporated. This is not the case with conventional casting, where design changes equate to expensive modifications to the tooling.

Comment 3. The captions of images are long. Rather than describing the image, they seem to discuss the image. I would recommend shortening them and focusing on the description of the image.

Response: Captions have been shortened per the reviewer's suggestion.

Comment 4. Simulation steps need to be explained further in the material and method section, or in a Supplementary Information document.

Response: Done. Please see also the response to **Reviewer 1 - comment 10**.

Comment 5. A colormap scale bar should be added to Fig. 6I.

Response: Done. Scale bar added.

Comment 6a. Finally, the readers may benefit from further explanations regarding the effect of residual stresses on distortion and mechanical properties.

Response: This is a good point; however, a detailed description is beyond the scope of this research article. References to publications dealing with distortion from residual stress in WAAM have been added to section 4.0 so that readers may pursue further details on this topic.

Comment 6b. A great extension of this work could be examining the structure under more realistic loading on the bones, such cyclic loading. I would recommend the authors to consider this in their future research.

Response: Thank you for your recommendation and constructive comments. We will keep that study plan for the future.

Reviewer #3

General comments (lettered A-E for clarity)

- A) The work is centered around production of a radial interface (i.e., bimetallic wall) in contrast to a planar interface resulting from a vertical/horizontal deposition of layers in the WAAM process.
- B) The technique presents an original approach from that aspect, and in fact, different WAAM techniques that produce bimetallic components have been reported previously which resulted in improving the overall mechanical performance (e.g., refs. [1,2]).
- C) While the authors' technique (radial bimetallic structure) presents a meaningful advance to the field (evident by the improved strength from the compression tests), I fail to see why it is of much significance than those other techniques already published in [1,2] which also provide additional strengths, albeit in other testing regimes.
- D) The radial bimetallic interface in this paper is indeed complex, however, there is no comparison against its planar counterparts to justify its significance in that sense.
- E) I have the following concerns/questions which I hope the authors can address:

Response: We appreciate this constructive feedback and hope the general comments are satisfactorily addressed. All subsequent numbered comments have been addressed, and a detailed response has been given.

- A) This work is a fundamental departure from producing planar interfaces and bimetallic walls. The concentric rings of material in this work form cylindrically arranged bimetallic interfaces, with resultant hoop stresses completely incongruous with the physics of linear wall deposition, bimetallic or otherwise. The technique relies upon two independent welding torches that deposit rapidly but follow unique deposition paths. A supplementary video has been uploaded that further highlights that deposition of the bimetallic couple occurs before either material has fully cooled. It is important to note that the planar bimetallic structures are typically made with one welding torch depositing one material on top of another.
- B) The suggested literature [1]: <https://doi.org/10.1016/j.msea.2022.143984>, [2]: <https://doi.org/10.1016/j.mtcomm.2022.104457>, are both studies that investigate laminated or alternating bimetallic layers. Augmentation of mechanical performance is indeed evidenced, but through an incomparable technique, see the response to **General comment A**.
- C) In any existing publication, the radial bimetallic technique promotes the interaction of bimetallic materials in ways unachievable by any type of linear or stratified deposition methodology. See also the response to **General comment A**. Concentric deposition combined with simultaneous solidification of dual materials goes beyond the simple strength improvements referenced and introduces the opportunity to investigate more complex properties such as corrosion resistance and fatigue life.
- D) Regarding significance, the comparison is appropriately drawn to linear deposition of any sort; see also the response to **General comment A**. Interface complexity is discovered in the 360 degrees of encircling material interaction between the casing and the core, totally uncharacteristic of the literature already cited in the manuscript and by the reviewer [1-4].
- E) The concerns are greatly appreciated and addressed as follows.

Comment 1. I am unable to support the authors' claim regarding the 'pre-tensioned concept' without in-depth microstructural analyses and discussion on and around the bimetallic interface (certainly EBSD and possibly TEM). Here, the paper can investigate the interface before and after compression to analyze the role of microstructure (crystallographic texture/grain morphology, etc.) on the attained mechanical strength. The EDS/EDX data which the authors provide can be expanded here also.

Response: EBSD analysis of the crystalline planes and texture have been pursued and added to the manuscript, with interface textures in a non-deformed bimetallic specimen shown in the updated manuscript Figure 6, or **Fig 3** below in this response document. The deformed samples did not allow us to polish to a level needed for EBSD analysis due to deformation at the interface.

Electron backscatter diffraction (EBSD) was used to characterize crystallite orientations with a scanning electron microscope (Apreo 2 FEG-SEM) and EDAX Velocity system. An accelerating voltage of 20 keV, stage tilt of 70 degrees, and 0.10 micron step size at a working distance of 8.6mm were used to acquire the Kikuchi patterns. TSL OIM software was used to provide analysis of the EBSD data, discarding data with a confidence index of less than 0.1. The EBSD data were not manipulated other than discarding the low confidence points after a Confidence Index standardization routine.

Close inspection of interface grain formation and crystallographic texture revealed strong ferritic orientation with $\{111\}$ poles aligned with the cylinder surface tangent direction for the mild steel casing, with an observable concentration of $\{101\}$ orientations in the immediate vicinity of the casing-core boundary (Fig. 6). Austenite formations were observed in the stainless core in the $\{101\}$ orientation.

Fig 3. (D) Location of SEM image and EBSD orientation maps observed in the mild steel casing (color keys indicate poles aligned with the cylinder tangent directions); (E) $\{001\}$, $\{011\}$, and $\{111\}$ pole figures for mild steel casing (normal direction is aligned with the cylinder tangent direction of the specimens); (F) Location of SEM image and EBSD orientation maps observed at the bimetallic interface between the core and casing; (G) $\{001\}$, $\{011\}$, and $\{111\}$ pole figures for ferrite grains at the bimetallic interface.

EBSB characterization of the crystalline texture at the interface found a surprising concentration of {111} fiber texture ferrite grain orientations aligned with the cylinder tangent direction in the mild steel, with traces of {101} orientations in the stainless steel. Very few {101} oriented grains were observed in the casing, which would have been anticipated due to alignment with the build direction. The effect of grain morphology on solidification residual stresses is of interest but considered beyond the scope of this article. Details for the further interested reader related to WAAM can be found in the references cited in the manuscript [44] – [46].

A third interface location (“Scan 18”), not detailed in the manuscript, is shown in **Fig 4** below as additional substantiation. When data in this scan are combined with the “Scan 13” interface analysis now detailed in the manuscript, the resultant harmonic texture maps (**Fig 5** and **Fig 6**) further strengthen confidence in the overall analysis. The findings of this additional interface location were consistent with those of the first interface location. The {111} fiber texture ferrite grain orientations aligned with the cylinder tangent direction in the mild steel are seen, with traces of {101} orientations again detected in the stainless steel. Very few {101} oriented grains were observed in the casing.

Fig 4. EBSD orientation maps observed at the bimetallic interface between the core and casing at a second interface location (Scan 18)

Fig 5. {001}, {011}, and {111} pole figures for ferrite grains at the bimetallic interface, using combined data from scans 13 and 18.

Fig 6. {001}, {011}, and {111} pole figures for ferrite grains at the bimetallic interface, combining data from scans 13 and 18.

A direct comparison against a sectioned post-compression-test bimetallic specimen was attempted, but the deformed specimen proved challenging to characterize and beyond the capability of the equipment currently available at our disposal at WSU.

Comment 2. The drilled bimetallic specimen exhibited a strength of 350 MPa, which is less than the ER70S-6 (370 MPa) and very close to the SS specimen (346 MPa) (page 13). Including a comparison with drilled ER70S-6 and drilled SS seems logical to me for a fair comparison.

Response: Clarifying details have been added to the manuscript regarding the purpose of the drilled relief passage; see also **Reviewer #1 - comment 6B** and its related manuscript edits. Briefly, the comparison between the drilled bimetallic specimen and the undrilled monolithic controls is intended to highlight the observation that the removal of the constraint inherent in a radially arranged bimetallic structure using a relief passage allows plastic deformation and returns the mechanical capability of the structure to that of a monolithic (undrilled) specimen. The implication is that the attribution of mechanical enhancement in the bimetallic specimen is appropriately drawn to the interaction of the casing and core as they restrict movement through residual pressure (See also the response to **comment 3** below).

Additional monolithic specimens in 308L stainless and mild steel were fabricated to support this response. These specimens were drilled centrally to form a relief passage, then tested in compression as suggested (**Fig 3** immediately following). The relief passage in the monolithic specimens predictably reduced the total loading experienced by the specimens. However, when the area of the drilled passage reduced the cross-sectional area used in yield stress calculations, yield stress was logically unchanged from the averages detailed in the manuscript for monolithic controls.

Figure 3: (A) Additional monolithic mild steel compression specimen prior to drilling; (B) Mild steel specimen with drilled passage; (C) Post-test monolithic drilled mild steel specimen; (D) Additional monolithic 308L compression specimen prior to drilling; (E) 308L specimen with drilled passage; (F) Post-test monolithic drilled 308L specimen.

Comment 3. While drilling the bimetallic specimen does relieve stress, it also provides more means for material to flow, hence, expectedly reduces strength. Please discuss this.

Response: Done. Additional discussion and clarifying details on the relief passage and material flow have been added to the discussion; see also Figure 6 and response to **comment 6b**.

Comment 4. It is essential to include the chemical composition of the two alloys in a stand-alone table.

Response: Done. Chemical compositions of the two alloys have been added.

Comment 5. Discuss the zig-zag feature of the bimetallic wall in the context of existing literature (e.g., [3] and a very recently published paper [4], of course, these references are not exhaustive)

Response: Done. The text has been added to section 4.0, citing the recommended literature and highlighting the fundamental difference between linear bimetallic interfaces and circular interfaces arranged radially. See also the response to **General comment A**.

Other minor comments

Comment 1. Introduction is too broad. I suggest gearing it a little bit for a slightly knowledgeable target audience to shorten it.

Response: Done. See also a response to **Reviewer #2 - comment 1**.

Comment 2. Panels E, G, M in Fig. 5 should be the star of the show, because they immediately show the benefit. Hence, I recommend making them the highlight of the figure.

Response: Done. A new figure was added, dedicated to highlighting the suggested images. Corresponding edits made to subsequent figure references.

Comment 3. Kindly be consistent with the terminology/acronyms (e.g., low carbon control vs. mild steel, 308SS vs. 308L).

Response: Done. Terminology and acronyms were corrected for consistency throughout the text.

Comment 4. Remove unnecessary repetition of the experiments or motivation (e.g., beginnings of chapters 3.0 and 4.0).

Response: Done. Repeated descriptions have been removed.

Comment 5. Use sequential referencing to figures and figure panels (i.e., Fig. 1a, Fig. 1b...etc. not Fig. 1a, Fig 4b, Fig. 1b, Fig. 2c), thereby making it easier for the reader to follow without flipping pages/scrolling many times).

Response: Done. The references to figures and figure panels have been updated to follow sequentially.

REVIEWERS' COMMENTS

Reviewer #1 (Remarks to the Author):

Dear Editor and authors,

I read all the comments one by one. All of my concerns have been addressed. There is no further comment. It is worthy of publication.

Thanks for the interesting work.

Reviewer #2 (Remarks to the Author):

The authors have adequately addressed all the reviewer's concerns; therefore, the revised manuscript can be accepted for publication.

Reviewer #3 (Remarks to the Author):

The authors have addressed my comments.